# Development of Porcine Monoclonal Antibodies with In Vitro Neutralizing Activity against Classical Swine Fever Virus from C-Strain E2-Specific Single B Cells

**DOI:** 10.3390/v15040863

**Published:** 2023-03-28

**Authors:** Lihua Wang, Rachel Madera, Yuzhen Li, Douglas P. Gladue, Manuel V. Borca, Michael T. McIntosh, Jishu Shi

**Affiliations:** 1Department of Anatomy and Physiology, College of Veterinary Medicine, Kansas State University, Manhattan, KS 66506, USA; rachelmadera@vet.k-state.edu (R.M.); yuzhen@vet.k-state.edu (Y.L.); 2Department of Agriculture, Agricultural Research Service, Plum Island Animal Disease Center, Greenport, NY 11944, USA; douglas.gladue@ars.usda.gov (D.P.G.); manuel.borca@ars.usda.gov (M.V.B.); 3Department of Molecular Genetics and Microbiology, University of Florida, Gainesville, FL 32611, USA; mmcintosh@peds.ufl.edu; 4Child Health Research Institute, Department of Pediatrics, University of Florida, Gainesville, FL 32611, USA

**Keywords:** classical swine fever (CSF), porcine, monoclonal antibody, neutralizing, C-strain, E2, single B cell

## Abstract

Neutralizing antibodies (nAbs) can be used before or after infection to prevent or treat viral diseases. However, there are few efficacious nAbs against classical swine fever virus (CSFV) that have been produced, especially the porcine-originated nAbs. In this study, we generated three porcine monoclonal antibodies (mAbs) with in vitro neutralizing activity against CSFV, aiming to facilitate the development of passive antibody vaccines or antiviral drugs against CSFV that offer the advantages of stability and low immunogenicity. Pigs were immunized with the C-strain E2 (CE2) subunit vaccine, KNB-E2. At 42 days post vaccination (DPV), CE2-specific single B cells were isolated via fluorescent-activated cell sorting (FACS) baited by Alexa Fluor™ 647-labeled CE2 (positive), goat anti-porcine IgG (H + L)-FITC antibody (positive), PE mouse anti-pig CD3ε (negative) and PE mouse anti-pig CD8a (negative). The full coding region of IgG heavy (H) chains and light (L) chains was amplified by reverse transcription-polymerase chain reaction (RT-PCR). Overall, we obtained 3 IgG H chains, 9 kappa L chains and 36 lambda L chains, which include three paired chains (two H + κ and one H + λ). CE2-specific mAbs were successfully expressed in 293T cells with the three paired chains. The mAbs exhibit potent neutralizing activity against CSFVs. They can protect ST cells from infections in vitro with potent IC_50_ values from 14.43 µg/mL to 25.98 µg/mL for the CSFV C-strain, and 27.66 µg/mL to 42.61 µg/mL for the CSFV Alfort strain. This study is the first report to describe the amplification of whole-porcine IgG genes from single B cells of KNB-E2-vaccinated pig. The method is versatile, sensitive, and reliable. The generated natural porcine nAbs can be used to develop long-acting and low-immunogenicity passive antibody vaccine or anti-CSFV agents for CSF control and prevention.

## 1. Introduction

Classical swine fever (CSF) is a highly contagious viral disease of domestic and wild pigs [1]. Pigs with CSF, also known as hog cholera, have clinical signs such as high fever, loss of appetite, lethargy, and a high mortality rate [2]. CSF is endemic to much of Asia, Central and South America, and parts of Europe and Africa [3,4]. For CSF-free countries, such as the United States, the disease represents a high-consequence threat to agriculture, security, and trade [5].

For CSF control and prevention, endemic countries mainly rely on routine vaccination with modified live virus vaccines (LAVs) [6,7]. However, such vaccines lack the ability of DIVA (Differentiation of Infected from Vaccinated Animals), which is essential to trade swine products with other countries. Without DIVA-compatible vaccines, CSF-free countries must rely on costly stamping-out policies, i.e., culling both vaccinated and infected herds in the event of an outbreak [1,7]. To date, alternative DIVA vaccine strategies based on CSFV envelope glycoprotein E2 have been developed. These include CSFV E2 subunit vaccines [8,9,10,11,12,13] and live chimeric virus vaccine made by replacing the E2 of the closely related bovine viral diarrhea virus with the E2 of CSFV [14,15,16]. Their DIVA concept is based on the fact that field-virus-infected animals can raise antibodies to an additional antigen (such as E^rns^ protein), while vaccinated animals only develop a CSFV E2-specific antibody response. Unfortunately, the E2-substituted chimeric virus vaccine (Suvaxyn CSF Marker) is only approved by the EU for emergency vaccination within restricted control. E2 subunit vaccines can effectively induce CSF protection, but they induced shorter immunity compared to the live attenuated vaccine, thus two or more doses are required followed by single-dose re-vaccination every six months [8,9,10,11,12]. These limitations indicate a need for development of new and better vaccine platforms for CSF control and prevention.

Neutralizing antibodies (nAbs) can directly neutralize pathogens and have been used to treat various viral infections in humans including Ebola virus, Human immunodeficiency virus, Influenza virus, Rabies virus, Respiratory syncytial virus, and SARS-CoV-2 virus [17,18,19,20,21]. They can be passively transferred into individuals before or after viral infection to prevent or treat diseases. Thus, there is a huge potential by using CSFV nAbs as passive antibody vaccines with DIVA capability to control and prevent CSF. In this study, we report the generation of natural porcine nAbs from single B cells of a CSFV C-strain E2 (CE2)-immunized pig, aiming to develop long-acting and low-immunogenicity passive antibody vaccines or anti-CSFV agents for CSF control and prevention.

## 2. Materials and Methods

### 2.1. Animals

Three conventional Large White–Duroc crossbred weaned specific-pathogen-free male piglets (3 weeks of age) were purchased from a commercial vendor. The pigs were fed with a standard commercial diet and housed in the Large Animal Research Center (LARC) at the Kansas State University. Animal care and protocols were approved by the Institutional Animal Care and Use Committee (IACUC#4212) at Kansas State University. All animal experiments were done under strict adherence to the IACUC protocol.

### 2.2. Cell Lines and Virus Strains

ST (Swine testicular cells, ATCC, CRL-1746) and 293T (human embryonic kidney cells, ATCC, CRL3216) cells were cultured in Dulbecco’s Modified Eagle’s Medium (DMEM, Gibco, Grand Island, NY, USA) supplemented with 10% fetal bovine serum (FBS, Atlanta Biologicals, Flowery Branch, GA, USA) and 1% penicillin-streptomycin solution (Gibco, NY, USA) at 37 °C in 5% CO_2_ incubator.

The CSFV C-strain (Chinese vaccine strain) and the CSFV Alfort strain are kept in our laboratory at Kansas State University.

### 2.3. Animal Immunization and E2 Antibody Testing

The CSFV E2 subunit vaccine KNB-E2 was prepared by simple hand mixing of purified CE2 with an oil-in-water emulsion adjuvant according to the procedure we described previously [9,22]. One dose (2 mL) KNB-E2 contains 100 μg of purified CE2 protein. Pigs were immunized with three doses intramuscularly with 2 mL of KNB-E2 at day 0, 14 and 28, respectively. Pigs were monitored daily. Sera were collected on day 0, 14, 28, and 35 days post vaccination (DPV).

The level of anti-E2 antibodies in collected serum samples was determined by using ELISA as described previously [9]. Briefly, 62.5 ng/mL of purified CE2 was used as a coating antigen on 96-well flat-bottomed microtiter plates (Corning, Corning, NY, USA). After three washes with PBS containing 0.05% Tween 20 (PBST), diluted sera were added to plates and incubated for 1 h at room temperature. Then, the ELISA plates were washed three times with PBST before detection of the CE2-specific antibodies by using horseradish peroxidase-conjugated goat anti-porcine IgG (Southern Biotech, Birmingham, AL, USA). The ELISA plates were developed using 3,3,5,5 tetramethylbenzidine (TMB) stabilized chromogen (Thermo Scientific, Waltham, MA, USA), followed by the addition of 2 N sulfuric acid to stop the reactions. Optical spectrophotometer readings at 450 nm were measured by using a SpectraMAX microplate reader (Molecular Devices, Silicon Valley, CA, USA) to determine the relative antibody concentrations in serum samples.

### 2.4. Cell Staining and Porcine Single B Cell Sorting

Blood samples were collected at 42 DPV from the pig with the highest level of CE2-specific antibody responses. Peripheral blood mononuclear cells (PBMCs) from whole blood were purified with Ficoll–Paque (1.077 g/L, Pharmacia, Upsala, Sweden) and SepMate™ PBMC isolation tubes (Stemcell Technologies, Vancouver, BC, Canada) according to the manufacturer’s instructions. After washing by fluorescent-activated cell sorting buffer (FACS buffer, PBS with 2% FBS), the PBMCs (1 × 10^7^ cells) were stained by adding staining cocktail (5 μg/mL per antibody or antigen) in FACS buffer containing Alexa Fluor™ 647-labeled CE2 (Prepared with Alexa Fluor™ 647 protein-labeling kit, Invitrogen, Waltham, MA, USA), goat anti-porcine IgG (H + L)-FITC antibody (Southern Biotech, AL, USA), PE mouse anti-pig CD3ε (BD Biosciences, Qume Dr San Jose, CA, USA), and PE mouse anti-pig CD8a (BD Biosciences, CA, USA). The cells and staining cocktail mixture were incubated in the dark on ice for 30 min. After washing two times with FACS buffer (400× *g*, 5 min per wash), the cells were re-suspended in 1 mL of pre-chilled FACS buffer and passed through a 70 μm cell strainer (BD Biosciences, CA, USA) prior to cell sorting. CE2- and IgG-specific (Alexa Fluor™ 647 and FITC positive, PE negative) single B cells were identified and sorted by a FACS Aria III cell sorter (BD Biosciences, CA, USA) at single-cell density into 96-well PCR plates containing 10 μL of buffer A: 1 μL 10xRT buffer from a High-Capacity cDNA Reverse Transcription Kit (Thermo Scientific, IL, USA), 2 μL 0.1 M Dithiothreitol (DTT, New England Biolabs, Ipswich, MA, USA) and 7 μL nuclease-free water (Invitrogen, CA, USA). The plates were stored at −80 °C until use.

### 2.5. Porcine IgG Gene-Specific Single B Cell RT-PCR

The plates containing single B cells were thawed on ice and spun briefly to collect liquid and cells in the bottom of wells. A volume of 5 μL of buffer B: 0.5 μL 10xRT buffer, 1 μL random primers, and 0.5 μL RNA inhibitor from High-Capacity cDNA Reverse Transcription Kit, 0.06 μL Igepal CA-630 (Sigma, Spruce St Saint Louis, MO, USA), and 2.94 μL nuclease-free water were added to each well. After incubation at 65 °C for 1 min, 55 °C for 30 s, 45 °C for 30 s, 35 °C for 30 s, 23 °C for 2 min in a thermocycler (BioRad T100, Hercules, CA, USA), 5 μL buffer C containing 0.5 μL 10xRT buffer, 0.8 μL dNTP Mix (100 mM), and 1 μL MultiScribe™ Reverse Transcriptase from High-Capacity cDNA Reverse Transcription Kit, and 2.7 μL nuclease-free water were added to each well. Reverse transcription was completed according to the following thermocycler conditions: 25 °C for 10 min, 37 °C for 120 min, 85 °C for 5 min and holding at 4 °C.

For designing PCR primers to amplify IgG gene chains from the cDNAs, porcine full-length IgG gene records were downloaded as plain text files from GenBank and aligned by using CLC Sequence Viewer 8.0 (Qiagen, Hilden, Germany). Degenerated primer sets (Table 1) were designed to amplify the whole coding regions of porcine IgG heavy (H) chain, kappa (κ) light (L) chain and lambda (λ) L chain. PCR reactions were set up with Phusion High-Fidelity DNA Polymerase (NEB, Ipswich, MA, USA): 10 μL 5× PhusionHF Buffer, 1 μL dNTPs (10 mM), 5 μL primer mix (2 μM of each), 5 μL cDNA, 1.5 μL DMSO, 0.5 μL Phusion DNA polymerase, and 27 μL nuclease-free water. PCR cycling was conducted at 98 °C for 30 s, 2 cycles of 98 °C for 10 s, 55 °C for 30 s, and 72 °C for 1 min 40 s (H chain) or 1 min (L chains), followed by 28 cycles of 98 °C for 10 s, 58 °C for 30 s, and 72 °C for 1 min 40 s (H chain) or 1 min (L chains), and the final extension was at 72 °C for 10 min. The PCR products were checked by 1% agarose gel electrophoresis.

### 2.6. Cloning and Expression of Porcine Monoclonal Antibodies in Mammalian Expression System

Paired amplicons (H and L chain PCR products) from the same well containing single B cells were purified with NucleoSpin Gel and a PCR clean-up kit (MACHEREY-NAGEL Inc., Allentown, PA, USA), digested by restriction enzymes EcoRI and XbaI, and cloned into a pcDNA3.1(+) mammalian expression vector (Invitrogen, CA, USA). The H and L chains which cloned into the vector were sequenced with the ABI 3730XL sequencer at a commercial company (MCLAB, South San Francisco, CA, USA). Alignments of nucleotide and deduced amino acid sequences were conducted using CLC Sequence Viewer 8.0 (Qiagen, Hilden, Germany).

To produce monoclonal antibodies (mAbs), 2 μg of each paired H and L chain constructs was co-transfected into 293T cells using TransFectin™ Lipid Reagent (BioRad, CA, USA) following the manufacturer’s instructions. After culturing for three days at 37 °C under 5% CO_2_ in 6-well plates, supernatants were harvested, aliquoted, and stored at −80 °C for later use.

### 2.7. Western Blotting and Indirect Immunofluorescence (IFA) Testing of Porcine Monoclonal Antibodies

For Western blotting, proteins (mAbs or CE2) were separated by electrophoresis in Mini-Protean TGX Gel (Bio-Rad, Hercules, CA, USA), and transferred to PVDF membranes (Millipore, Burlington, MA, USA). Membranes were blocked with 5% milk, and then incubated with primary antibodies. Incubation with horseradish peroxidase (HRP)-conjugated secondary antibodies, detection and imaging were performed as we described previously [9].

For testing the neutralizing activity of mAbs against CSFVs, supernatants containing porcine mAbs were purified by a HiTrap™ Protein G column (GE Healthcare Life Sciences, Norristown, PA, USA), and the concentrations were measured using a BCA assay kit (Pierce, Appleton, WI, USA) according to the manufacturer’s recommendations. The purified mAbs were serially diluted two-fold in 96-well plates. The diluted samples (in duplicate) were incubated with 100 TCID_50_ (50% tissue culture infective dose) of the CSFV C-strain or the Alfort stain in DMEM with 2% FBS for 1 h at 37 °C. Residual virus infectivity was determined by adding 1 × 10^4^ ST cells to each well with mAb-virus mixture in 96-well plates and incubated at 37 °C for 4 days. The cells were subjected to immunofluorescence staining with E2-specific mAb WH211 (Animal and Plant Health Laboratories Agency, Wey Bridge, UK) and Alexa Fluor 488 goat anti-mouse IgG (H  +  L) (Life Technologies, Carlsbad, CA, USA). Finally, the plate was washed three times with PBST and examined under a fluorescence microscope (EVOS FLc, Waltham, WA, USA). The calculation of IC_50_ (half maximal inhibitory concentration) value was carried out by the web-based tool (https://www.aatbio.com/tools/ic50-calculator, accessed on 13 March 2023).

## 3. Results

### 3.1. Isolation of CSFV C-Strain E2 Protein-Induced Single Porcine B Cells

We vaccinated the pigs with three doses of KNB-E2, aiming to induce strong anti-CE2 immune responses and a high proportion of CE2-specific B cells in their peripheral blood. As we expected, the pigs generated a very high level of antibody responses against CE2 (Figure 1A) after 14 DPV. Pig #1, which showed the earliest and the highest level of CE2-specific antibody responses in the same group, was selected for isolation of PBMCs. CE2^+^IgG^+^ B cells in porcine PBMCs were determined based on the fluorescent signals of CE2-Alexa Fluor™ 647 and anti-porcine IgG (H + L)-FITC. Anti-pig CD3ε-PE and anti-pig CD8a-PE were used to distinguish the CD4^+^ T lymphocytes, CD8^+^ T lymphocytes, thymocytes, and natural killer (NK) cells. Through sorting analysis, approximately 2% of isolated PBMCs were identified with the phenotype of Alexa Fluor™ 647^+^ FITC^+^ PE^-^ (Figure 1B), which were gated and sorted at single-cell density into a 96-well PCR plate for downstream single B cell IgG gene-specific RT-PCR reactions.

### 3.2. Amplification of IgG Heavy and Light Chains from Porcine Single B Cell

To efficiently amplify the whole coding regions of porcine IgG chains, we designed degenerate primers based on the available full-length porcine IgG genes in GenBank (Figure 2A). For conveniently cloning the amplified products into pcDNA3.1(+) vectors and efficiently expressing in mammalian cell culture, all forward primers include a EcoRI restriction enzyme site, a kozak sequence, and a translation initiation codon. All reverse primers include a XbaI restriction enzyme site, and a translation stop codon (Table 1). The primers worked well with our cell lysis and RT-PCR systems. PCR products showed the expected size via agarose gel electrophoresis analysis (Figure 2B). From 384 sorted single B cells, we successfully obtained 3 IgG H chains, 9 κ chains and 36 λ chains, which include three paired chains (H1 + λ1, H9 + κ9, and H11 + κ11).

### 3.3. Genetic Characterizon of the Paired Chains Recovered from CE2^+^IgG^+^ Porcine Single B Cells

All three paired chains were cloned into pcDNA3.1(+) mammalian expression vectors, and their sequences were obtained by sequencing. The coding region of H1 is 1422 bp (base pair), which showed 91.5% nucleic acid (nt) identity with H9 (1, 404 bp), and 91.4% nt identity with H11 (1, 404 bp). H9 and H11 showed 99.9% nt identity. The coding region of L chain λ1 is 723 bp length. The coding region of L chain κ9 (708 bp) showed 51.2% nt identity with κ11 (705 bp) (Figure 3).

H1, H9 and H11 encode polypeptides with 473 amino acids (aa), 467aa, and 467 aa, respectively. H1 showed 27.9% aa identity with H9 and H11. H9 showed 99.7% aa identity with H11. They consist of similar components with the reported porcine IgG H chains including the variable domain (VH), hinge region, constant domains (CH1, CH2, and CH3), conserved residues (cysteine, and tryptophan), and N-linked glycosylation site in the CH2 domain. Interestingly, H1 encodes a distinct hinge region with other H chains, whereas H9 and H11 have the same hinge-coding sequences with IgG4 subclasses (Figure 4).

By comparing the deduced amino acid sequences of constant regions of H1, H9 and H11 with known pig IgG subclasses (IgG1, IgG2a, IgG2b, IgG2c, IgG3, IgG4, IgG5a, IgG5b, and IgG5c) [23], we found that H1 showed the highest identity (92%) with IgG1, and the other two chains (H9 and H11) showed the highest identity (95%) with IgG4 (Table 2).

### 3.4. The Porcine mAbs Can Specifically Bind to CSFV CE2 Protein

All three paired chains (H1 + λ1, H9 + κ9, and H11 + κ11) are successfully expressed after transfecting them into 293T cells (Figure 5A). As expected, the weight of L chains is approximately 25 kDa, and the weight of H chains is approximately 50 kDa. The produced mAbs (mAb1, mAb9, and mAb11) in the collected supernatants can specifically react with CE2 protein (Figure 5B). In addition, all mAbs can recognize both native and reduced CE2, which indicates that the expressed porcine mAbs recognize linear epitopes of CE2.

### 3.5. The Porcine mAbs Exhibited High Potency in Neutralizing CSFVs

We evaluated the neutralizing activity of purified porcine mAbs against the CSFV C-strain (vaccine strain) and the CSFV Alfort strain (a highly virulent strain) in ST cells via IFA. As shown in Figure 6A, all three mAbs can 100% protect ST cells (no fluorescence signal) from infection of the CSFV C-strain at 62.5 µg/mL. The IC_50_ values of mAb1, mAb9, and mAb11 are 21.22 µg/mL, 25.98 µg/mL, and 14.43 µg/mL, respectively. For CSFV Alfort stain, all three mAbs can 100% protect ST cells from infection at 125 µg/mL. The IC_50_ values of mAb1, mAb9, and mAb11 are 35.64 µg/mL, 42.61 µg/mL, and 27.66 µg/mL, respectively.

## 4. Discussion

As described in the introduction, nAbs have a huge potential to be used as passive antibody vaccines with DIVA capability to control and prevent CSF. We have successfully generated a mouse nAb (6B211) from CE2-immunized mouse, which showed potent neutralizing activity against CSFVs [24]. However, mouse-originated nAbs that are produced by hybridoma technology can induce rapid anti-mouse antibody responses in pigs, which will hasten the clearance of the mouse nAbs and produce undesirable allergic reactions [19,25,26,27]. Natural nAbs from pigs can avoid these disadvantages, and can have a lower risk of immunogenicity and longer half-life.

In this study, we tried to generate CSFV nAbs directly from pigs with the single B cell RT-PCR technology. B cells are the main source for obtaining antigen-specific antibody sequences [28]. Single B cell isolation following direct gene amplification and transient expression in animal cells for screening has been proven to be a powerful strategy to obtain natural mAbs [28,29,30,31,32,33]. Selection of immunization antigen and identifying antigen-specific B cells are two critical components of this strategy. The immunized antigen should have the ability to induce strong immune responses, especially nAb responses. The glycoprotein E2 of CSFV exposed on the outer surface is responsible for inducing nAbs and eliciting protective immunity against CSFV [4,8,9,10,11,12,13]. Thus, we administered three doses of CE2 based vaccine to induce a high proportion of nAb-secreting B cells in peripheral blood of immunized pigs. Indeed, a large portion (2%) of PBMCs were identified with the E2-specific phenotype when checked by FACS (Figure 1B). Using FACS sorting with fluorochrome-conjugated antigen (Alexa Fluor™ 647-CE2) and anti-pig IgG secondary antibody (FITC-anti-pig IgG), we successfully isolated CE2- and IgG-specific B cells from the pig PBMCs. Through RT-PCR with the porcine IgG gene-specific primer sets designed in this study, we obtained 3 IgG H chains, 9 κ chains and 36 λ chains from 384 sorted single B cells, which indicate that the combination of antigen and anti-pig IgG secondary antibody is sufficient to capture CE2-specific B cells.

We generated three mAbs (mAb1, mAb9, and mAb11) by using the paired chains recovered in this study. The mAbs can specifically bind CE2, and can neutralize the CSFV C-strain and the CSFV Alfort strain in vitro. They can protect ST cells from infections in vitro with potent IC_50_ values from 14.43 µg/mL to 25.98 µg/mL for the CSFV C-strain, and 27.66 µg/mL to 42.61 µg/mL for the CSFV Alfort strain. By comparing amino acid sequences of their constant regions with known pig IgG subclasses, we found that mAb1 belongs to IgG1, and the other two mAbs (mAb9 and mAb11) belong to IgG4. IgG1 and IgG4 have longer circulating-life than other IgG subtypes [34]. In addition, IgG1 is the principal IgG to cross the placenta [35]. Thus, the porcine-originated mAbs obtained in this study are good candidates for developing long-acting and low-immunogenicity passive antibody vaccine or anti-CSFV agents for CSF control and prevention, which can benefit both neonatal and adult pigs. Further analysis of their affinity to CSFV E2 proteins, their ability to neutralize other CSFVs, and the ability to protect pigs against challenge of virulent CSFVs in vivo will be performed and published separately at a later date.

## 5. Conclusions

In this study, we generated CSFV nAbs directly from pigs by using the single B cell RT-PCR technology. The method is versatile, sensitive, and reliable. The generated natural porcine nAbs can be used to develop long-acting and low-immunogenicity passive antibody vaccine or anti-CSFV agents for CSF control and prevention.

## Figures and Tables

**Figure 1 viruses-15-00863-f001:**
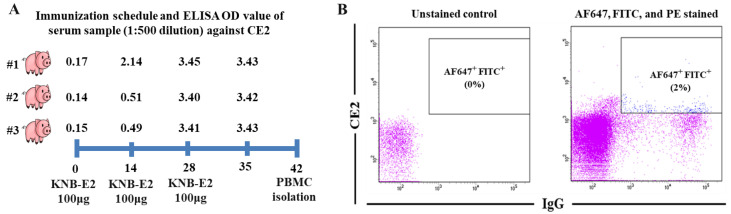
Isolation of CE2^+^IgG^+^ porcine B cells. (**A**) Immunization schedule and anti-CE2 antibody titers in serum samples collected at 0, 14, 28, and 35 DPV. (**B**) Distribution and proportion of CE2^+^IgG^+^ (AF647^+^FITC^+^) porcine B cells in peripheral blood of pig #1 as determined by FACS.

**Figure 2 viruses-15-00863-f002:**
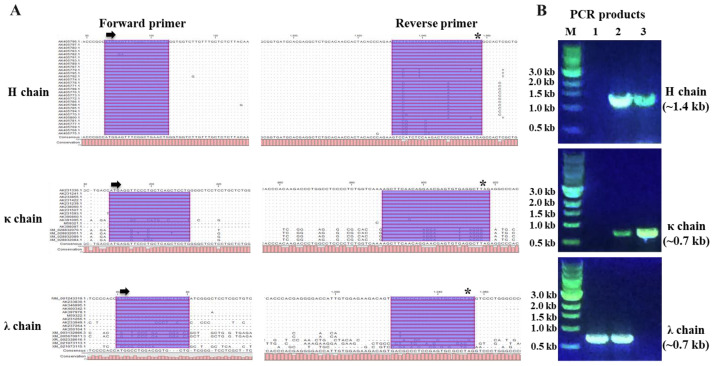
Primer design and single-cell RT-PCR to amplify whole coding regions of porcine IgG chains. (**A**) The 5′ end and 3′ end alignment of porcine IgG heave and light chain genes. Arrow indicates the translation initiation codon. Asterisk indicates the translation stop codon. (**B**) Representative 1% gel electrophoresis patterns of the PCR products. M, marker; 1-3, PCR products from different single B cells.

**Figure 3 viruses-15-00863-f003:**
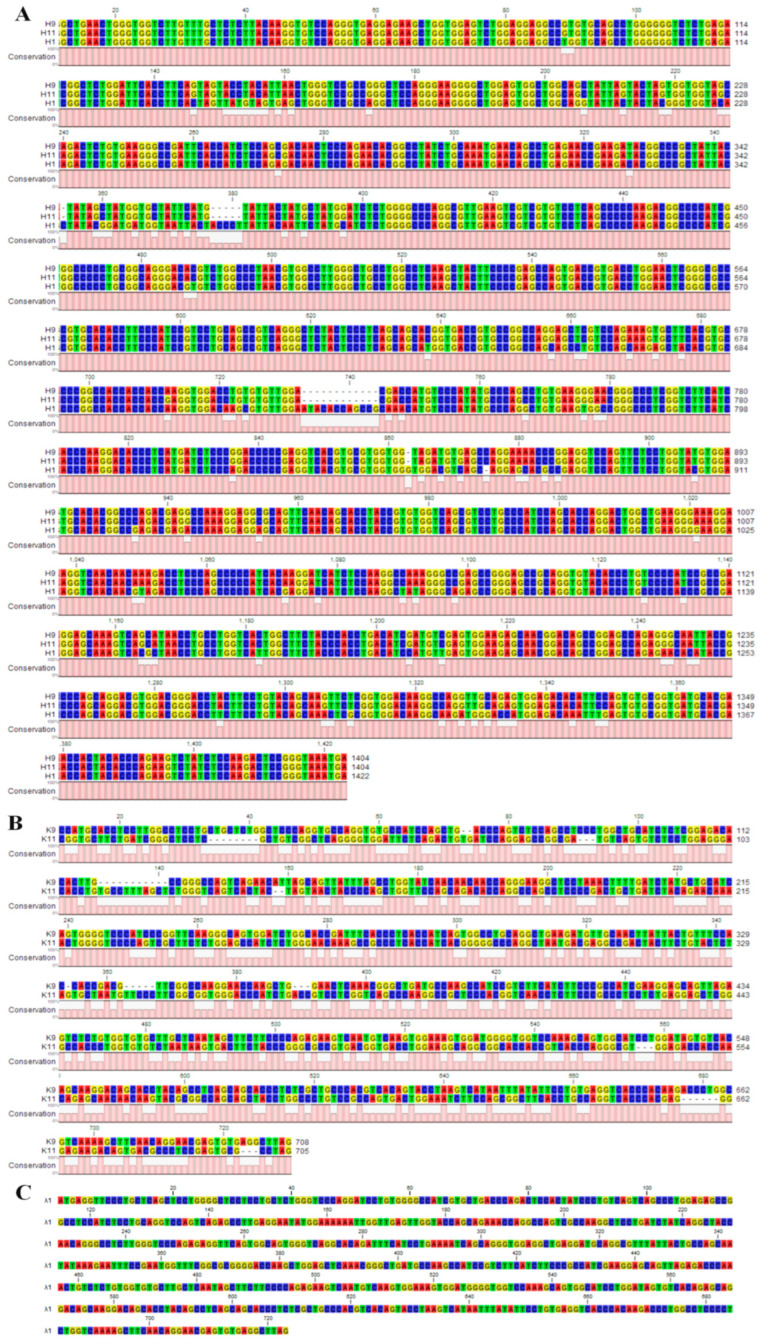
Sequence and sequence alignment of the paired chains from CE2^+^IgG^+^ porcine single B cells. (**A**) Sequence and sequence alignment of H1, H9 and H11. (**B**) Sequence and sequence alignment of L chains κ9 and κ11. (**C**) Sequence of L chain λ1.

**Figure 4 viruses-15-00863-f004:**
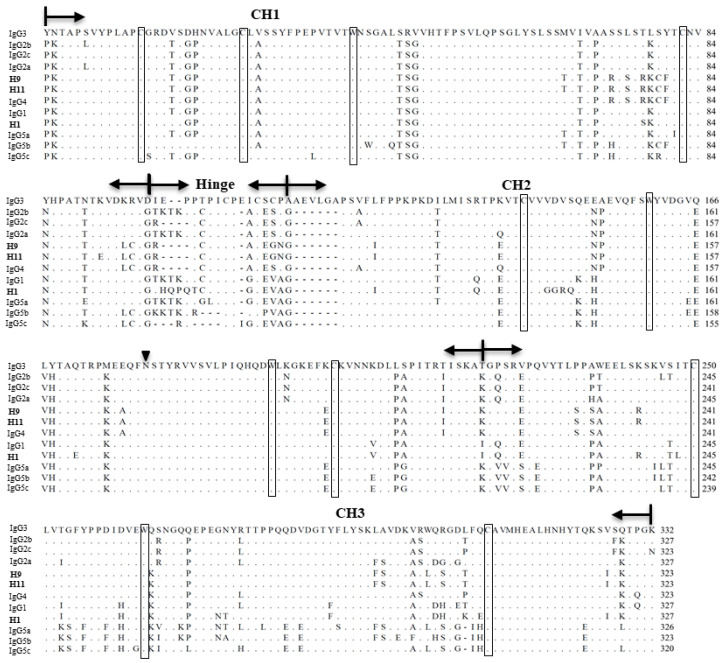
Comparison of the deduced amino acid sequences of H1, H9 and H11 with pig IgG subclasses. Dots indicate the same sequence and dashes indicate deletions. The conserved cysteines and tryptophans are boxed. An inverted triangle marks the conserved N-linked glycosylation site. Sequences of IgG subclasses were obtained from previous published data [23].

**Figure 5 viruses-15-00863-f005:**
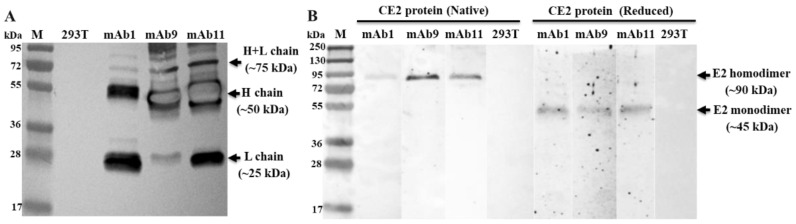
Analysis of the expression and CE2-binding activity of porcine mAbs by Western blotting. (**A**) Expression of porcine mAbs in 293T cell. Supernatants from H1 + λ1 (mAb1), H9 + κ9 (mAb9), H11 + κ11 (mAb11) transfected 293T cells were loaded on Mini-Protean TGX Gel (Bio-Rad, CA, USA). Supernatant from non-transfected 293T cell was used as control. The proteins were then transferred to PVDF membrane. The membrane were blocked and then incubated with anti-pig IgG (Sigma, Saint Louis, MO, USA). (**B**) Porcine mAbs can bind CE2 protein. Purified CE2 proteins were treated without (Native) or with β-mercaptoethanol (Reduced) and separated by SDS-PAGE in a Mini-Protean TGX Gel. The proteins were then transferred to PVDF membrane. The membranes were blocked and then incubated with supernatants of mAb1, mAb9, mAb11, and 293T cell, respectively. M, moledular weight marker.

**Figure 6 viruses-15-00863-f006:**
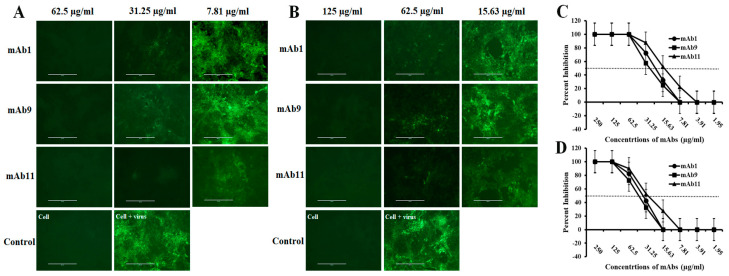
IFA testing the neutralizing activity of porcine mAbs (mAb1, mAb9, mAb11) against CSFVs. (**A**) IFA testing the neutralizing activity of porcine mAbs against the CSFV C-strain. (**B**) IFA testing the neutralizing activity of porcine mAbs against the CSFV Alfort strain. 4 days post infection (DPI). No green fluorescent signal means 100% inhibition. Scale bar: 200 μm. (**C**) Percentage inhibition of porcine mAbs against the CSFV C-strain. (**D**) Percentage inhibition of porcine mAbs against the CSFV Alfort strain. The dotted line represents IC_50_ (half maximal inhibitory concentration) value.

**Table 1 viruses-15-00863-t001:** Primer sets to amplify the whole coding regions of porcine IgG chains.

Primer Name	Sequence (5′-3′)	PCR Product
D-WHF1	CGGAATTCGCCATGGAGTTTCGGCTGAACTG	whole coding region of porcine IgH chain
D-WHF2	CGGAATTCGCCATGGGATTTCGGCTGAACTG	
D-WHR1	GCTCTAGATCATTTACCCGGAGTCTTGGAGATAGAC	
D-WHR2	GCTCTAGATCATTTACCCKGAGTCTKGRAGAYGGAC	
D-WκF1	CGGAATTCACCATGAGGTTCCCTGCTCAGCTCCTG	whole coding region of porcine Igκ chain
D-WκF2	CGGAATTCACCATGAGGGCCCCCATGCACCTCCTTG	
D-WκF3	CGGAATTCACCATGAGGGTCCCCGCTCAGCTCCTG	
D-WκR1	GCTCTAGACTAAGCCTCACACTCGTTCCTGYTGAAGCT	
D-WκR2	GCTCTAGACTAACACTCTCCTCTGTTGAAGCTCTTGG	
D-WλF1	CGGAATTCACCATGGCCTGGAYCCCTCTCCT	whole coding region of porcine Igλ chain
D-WλF2	CGGAATTCACCATGGCCTGGACGGTGCTTCTG	
D-WλF3	CGGAATTCACCATGAGGCCCAGGTCAGGCCAG	
D-WλR1	GCTCTAGACTAGGCGCACTCGGAGGGCRT	

Note: Degenerate bases were included in these sequences, including K = G or T, Y = C or T, R = A or G. The restriction enzyme sites of EcoRI and XbaI are underlined.

**Table 2 viruses-15-00863-t002:** Amino acid sequence identity matrix of H1, H9 and H11 constant regions with known pig IgG subclasses.

Subclasses	IgG3	H9	H11	IgG4	IgG2b	IgG2c	IgG2a	IgG1	H1	IgG5a	IgG5b	IgG5c
IgG3	ID	0.80	0.80	0.81	0.83	0.83	0.82	0.82	0.79	0.75	0.75	0.75
**H9**	0.80	ID	1.00	**0.95**	0.88	0.90	0.89	0.87	0.85	0.82	0.81	0.81
**H11**	0.80	1.00	ID	**0.95**	0.88	0.89	0.89	0.87	0.85	0.82	0.80	0.81
IgG4	0.81	0.95	0.95	ID	0.91	0.94	0.91	0.88	0.84	0.80	0.79	0.81
IgG2b	0.83	0.88	0.88	0.91	ID	0.97	0.96	0.92	0.86	0.84	0.80	0.81
IgG2c	0.83	0.90	0.89	0.94	0.97	ID	0.94	0.90	0.86	0.82	0.78	0.81
IgG2a	0.82	0.89	0.89	0.91	0.96	0.94	ID	0.92	0.86	0.85	0.81	0.81
IgG1	0.82	0.87	0.87	0.88	0.92	0.90	0.92	ID	0.92	0.87	0.83	0.84
**H1**	0.79	0.85	0.85	0.84	0.86	0.86	0.86	**0.92**	ID	0.82	0.80	0.80
IgG5a	0.75	0.82	0.82	0.80	0.84	0.82	0.85	0.87	0.82	ID	0.90	0.88
IgG5b	0.75	0.81	0.80	0.79	0.80	0.78	0.81	0.83	0.80	0.90	ID	0.89
IgG5C	0.75	0.81	0.81	0.81	0.81	0.81	0.81	0.84	0.80	0.88	0.89	ID

Note: Amino acid sequences of IgG subclass were obtained from previous published data [23]. Those with the highest identity of H1, H9 and H11 with known pig IgG subclasses are shown in bold.

## Data Availability

The datasets for the current study are available from the corresponding author on reasonable request.

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
