# Peer review of "Development of Porcine Monoclonal Antibodies with In Vitro Neutralizing Activity against Classical Swine Fever Virus from C-Strain E2-Specific Single B Cells"

_viruses, 2023, doi:10.3390/v15040863_

Round 1
Reviewer 1 Report
The manuscript reported three porcine monoclonal antibodies (mAbs) with neutralizing activity against classical swine fever virus (CSFV) developed from the C-strain E2 specific single B cells, which would be useful for CSF treatment and prevention. There are some concerns to be answered before accepted for publication.
1. The porcine mAbs isolated from the C-strain E2 immunized pigs showed well neutralizing activity to C-strain and Alfort strain. However, no data are provided for the sensitivity and specificity of the mAbs. The authors need to test more CSFV isolates of different genotypes as well as the other related viruses to determine their specificity and virus recognition spectrum.
2. The porcine mAbs expressed in 293T cells are suggested to be purified from the cultured supernatants to determine their expression quantity. Furthermore, affinity of the mAbs to E2 protein should be determined using the purified mAbs, and compared to that of the murine mAbs.
3. As the 3 porcine mAbs recognize linear epitopes of E2 protein, the epitopes recognized by the mAbs are suggested to be identified to determine whether they recognize the same epitope or not.
4. In Figure 6, the negative or virus controls without mAbs should be added to determine the inhibition efficiency of mAbs to CSFV infections.
Author Response
We would like to thank you for your thoughtful comments and efforts. These comments help us a lot for improving our manuscript. Following are a point-by-point response to the comments.
General comments
The manuscript reported three porcine monoclonal antibodies (mAbs) with neutralizing activity against classical swine fever virus (CSFV) developed from the C-strain E2 specific single B cells, which would be useful for CSF treatment and prevention.
Response: We would like to thank the reviewer for this positive evaluation.
Specific comments
- The porcine mAbs isolated from the C-strain E2 immunized pigs showed well neutralizing activity to C-strain and Alfort strain. However, no data are provided for the sensitivity and specificity of the mAbs. The authors need to test more CSFV isolates of different genotypes as well as the other related viruses to determine their specificity and virus recognition spectrum.
Response: We would like to thank for the valuable suggestions! This manuscript is mainly focus on describing the amplification of whole-porcine IgG genes from single B cells of CSFV C-strain E2 vaccinated pig and the confirmation of neutralizing activities of the expressed monoclonal antibodies from 293T cells. We will expand our study to test their sensitivity, specificity and virus recognition spectrum, and we will report the results in the follow-up papers.
- The porcine mAbs expressed in 293T cells are suggested to be purified from the cultured supernatants to determine their expression quantity. Furthermore, affinity of the mAbs to E2 protein should be determined using the purified mAbs, and compared to that of the murine mAbs.
Response: Thanks for this thoughtful suggestion! We will perform further analysis and characterization of these porcine mAbs, which will include their reactivity and affinity to E2 proteins from different CSFV genotypes, and comparison to that of murine anti-C-strain E2 mAbs generated in our lab (Wang, L. et al. BMC Vet. Res. 2020, 16(1):14; Mi, S., et al. Front. Immunol. 2022, 13:930631). The results will be reported in our follow-up papers.
- As the 3 porcine mAbs recognize linear epitopes of E2 protein, the epitopes recognized by the mAbs are suggested to be identified to determine whether they recognize the same epitope or not.
Response: We will perform further characterization of these porcine mAbs, which will include identifying their epitopes on C-strain E2 protein. We will publish the results in the follow-up paper as soon as possible.
- In Figure 6, the negative or virus controls without mAbs should be added to determine the inhibition efficiency of mAbs to CSFV infections.
Response: We added the cell control (without virus and mAbs) and virus control (without mAbs) in Figure 6.
Reviewer 2 Report
Summary of the paper:
In the paper by Wang, L, et. , the authors’ goal is to identify suitable neutralizing antibodies from pig to utilize for CSF control and prevent, as the current methods are insufficient in the area of differentiation of infected from vaccinated. The authors vaccinate healthy pigs with the E2 strain to obtain an antibody response. The antibodies were collected from various days post vaccination, sorted and sequenced. The authors expressed the three pairs in 293T cells and confirmed the neutralizing ability at dilution of up to 1L80 with CSVF in ST cells. The paper is well written, and the authors included the necessary controls for the experiments.
Questions to the authors:
1) The discussion section would benefit from
a. Discussion on the binding affinity of each antibody as shown on the western blot. It seems that mAB1 has the least binding a activity against CE2 protein, was a loading issue?
b. Discussion can also use some explanation as to the upper limit of this methodology, seeing the neutralizing effect disappear at larger dilutions would put the efficiency of this long-term method into question
c. PBMCs was used as early as the methods section but n ever defined.
2) The title of the paper could add “in-vitro”, the mAb were not tested in the pigs and the title could be misleading
Author Response
We would like to thank you for your thoughtful comments and efforts. These comments help us a lot for improving our manuscript. Following are a point-by-point response to the comments.
General comments
In the paper by Wang, L, et. al, the authors’ goal is to identify suitable neutralizing antibodies from pig to utilize for CSF control and prevent, as the current methods are insufficient in the area of differentiation of infected from vaccinated. The authors vaccinate healthy pigs with the E2 strain to obtain an antibody response. The antibodies were collected from various days post vaccination, sorted and sequenced. The authors expressed the three pairs in 293T cells and confirmed the neutralizing ability at dilution of up to 1:80 with CSVF in ST cells. The paper is well written, and the authors included the necessary controls for the experiments.
Response: We would like to thank the reviewer for this positive evaluation.
Specific comments
- The discussion section would benefit from a) Discussion on the binding affinity of each antibody as shown on the western blot. It seems that mAb1 has the least binding activity against CE2 protein, was a loading issue? b) Discussion can also use some explanation as to the upper limit of this methodology, seeing the neutralizing effect disappear at larger dilutions would put the efficiency of this long-term method into question. c) PBMCs was used as early as the methods section but never defined.
Response: Thanks for the valuable suggestions!
a) We loaded equal volume of supernatants for Western-blotting analysis. mAb1 showed lower binding activity against CE2 protein than other two mAbs (Figure 6B) may be caused by its lower expression level in 293T cells (Figure 6A). We added this explanation this in discussion section.
b) We evaluated neutralizing activities of the porcine mAbs in the cultured supernatants against CSFV C-strain (vaccine strain) and CSFV Alfort strain (highly virulent strain) in this study. We will purify the mAbs from supernatants and expand our study to test their affinity to CSFV E2 proteins, their ability to neutralize other CSFVs, and the ability to protect pigs against challenge of virulent CSFVs in vivo. These results will be published in the follow-up papers.
c) PBMCs has been defined in the methods section.
- The title of the paper could add “in-vitro”, the mAb were not tested in the pigs and the title could be misleading.
Response: We added in vitro to the title of this manuscript. The title has been changed to “Development of porcine monoclonal antibodies with in vitro neutralizing activity against classical swine fever virus from C-strain E2 specific single B cells”.
Reviewer 3 Report
The author developed 3 porcine mAbs from E2-vaccinated pigs, which showed potent neutralization to CSFVs in in-vitro research. It would be interesting to see some results from the animal experiment of the porcine mAbs in future research.
The overall quality of the research is good. However, some arguments and references in the paper can be updated. In line50-60, the author listed some limitations of E2 vaccination. However, recent studies showed the sound effeteness of some newly marketed E2 vaccines in China.
Author Response
Dear Reviewer,
We would like to thank you for your positive evaluation of our manuscript! In the revised manuscript, we updated the description and references about E2 subunit vaccine. The descriptions have been changed to “Unfortunately, the E2 substituted chimeric virus vaccine (Suvaxyn CSF Marker) is approved by EU only for emergency vaccination within restricted control. E2 subunit vaccines can effectively induce CSF protection, but they induced shorter immunity compared to the live attenuated vaccine, thus two or more doses are required followed by single-dose re-vaccination every six months [8-12]”. In addition, we purified the porcine monoclonal antibodies from the supernatants and retested in vitro neutralizing activities of these mAbs against CSFVs (C-strain and Alfort strain). The results showed that they can protect ST cells from infections in vitro with potent IC50 values from 14.43 µg/ml to 25.98 µg/ml for CSFV C-strain, and 27.66 µg/ml to 42.61 µg/ml for CSFV Alfort strain. We updated the Figure 3 and Figure 6 as well. We hope the manuscript after careful revisions meet your expectations.
Thank you for your thoughtful comments and efforts!
Best regards,
Lihua Wang
Reviewer 4 Report
The manuscript by wang et al describes three porcine monoclonal antibodies could neutralize classical swine fever virus and the mAbs can potently neutralize CSFVs. Overall, the study is well design and merit to be published on Viruses. However, I have several concerns,
Minor concerns.
1. Why the authors do not use the purified antibodies to perform the naturalization test? Then we can know the antibody concentrations.
2. Figure 3 is not clear, the resolution ratio is too low to see the alignment results, promote it.
3. H9 and H11 showed 99.9% nt identity, how about the aa sequence? The authors used H9 + κ9 and H11 + κ11 to expression antibodies, does the authors compered the H9 +κ11 or H11 + κ9 with H9 + κ9 and H11 + κ11?
Author Response
Dear Reviewer,
We would like to thank you for your positive evaluation of our manuscript! In the currently revised version, we purified the porcine monoclonal antibodies from the supernatants and retested in vitro neutralizing activities of these mAbs against CSFVs (C-strain and Alfort strain). The results showed that they can protect ST cells from infections in vitro with potent IC50 values from 14.43 µg/ml to 25.98 µg/ml for CSFV C-strain, and 27.66 µg/ml to 42.61 µg/ml for CSFV Alfort strain. We improved the resolution ratio and quality of Figure 3. We updated the description in Results 3.3 as “H1, H9 and H11 encode polypeptides with 473 amino acids (aa), 467aa, and 467 aa, respectively. H1 showed 27.9 % aa identity with H9 and H11. H9 showed 99.7% aa identity with H11.” It could be very interesting research to compare the original porcine mAbs to hybrid mAbs which contain heavy chains and light chains from different single B cells. We would like to test them and report the results in the follow-up papers. We hope the manuscript after careful revisions meet your expectations!
Thanks for your thoughtful comments and efforts!
Best regards,
Lihua Wang
Round 2
Reviewer 1 Report
I am regreted that the authors do not add any data for the monoclonal antibodies I suggested.
Author Response
Dear Reviewer,
We are sorry to hear that our previously revised manuscript has not met your expectations. In the currently revised version, we purified the porcine monoclonal antibodies from the supernatants and retested in vitro neutralizing activities of these mAbs against CSFVs (C-strain and Alfort strain). The results showed that they can protect ST cells from infections in vitro with potent IC50 values from 14.43 µg/ml to 25.98 µg/ml for CSFV C-strain, and 27.66 µg/ml to 42.61 µg/ml for CSFV Alfort strain. We updated description and reference about E2 subunit vaccine in the introduction part. We added the data of amino acid identities of Heavy chains. In addition, we updated the Figure 3 and Figure 6. We hope the manuscript after careful revisions meet your expectations.
Thanks for your thoughtful comments and efforts!
Best regards,
Lihua Wang
Round 3
Reviewer 1 Report
OK.